# Simplified Determination of RHEED Patterns and Its Explanation Shown with the Use of 3D Computer Graphics

**DOI:** 10.3390/ma14113056

**Published:** 2021-06-03

**Authors:** Łukasz Kokosza, Jakub Pawlak, Zbigniew Mitura, Marek Przybylski

**Affiliations:** 1Faculty of Metals Engineering and Industrial Computer Science, AGH University of Science and Technology, al. Mickiewicza 30, 30-059 Kraków, Poland; mitura@agh.edu.pl; 2Faculty of Physics and Applied Computer Science, AGH University of Science and Technology, al. Mickiewicza 30, 30-059 Kraków, Poland; jakub.pawlak@agh.edu.pl (J.P.); marprzyb@agh.edu.pl (M.P.); 3Academic Centre for Materials and Nanotechnology, AGH University of Science and Technology, al. Mickiewicza 30, 30-059 Kraków, Poland

**Keywords:** nanostructured materials, kinematical diffraction theory, Ewald construction, computer visualization

## Abstract

The process of preparation of nanostructured thin films in high vacuum can be monitored with the help of reflection high energy diffraction (RHEED). However, RHEED patterns, both observed or recorded, need to be interpreted. The simplest approaches are based on carrying out the Ewald construction for a set of rods perpendicular to the crystal surface. This article describes how the utilization of computer graphics may be useful for realistic reproduction of experimental conditions, and then for carrying out the Ewald construction in a reciprocal 3D space. The computer software was prepared in the Java programing language. The software can be used to interpret real diffractions patterns for relatively flat surfaces, and thus it may be helpful in broad research practice.

## 1. Introduction

High-quality thin films and related structures can be obtained using molecular beam epitaxy (e.g., [1,2,3]) or pulsed laser deposition (e.g., [4,5,6,7]). The ordering of atoms at the surface of samples can be verified by a number of different characterization techniques. One of them is reflection high-energy electron diffraction (RHEED), employed to monitor changes at surfaces of films grown both by molecular beam epitaxy and pulsed laser deposition. RHEED patterns may be quite complicated, in particular for disordered surfaces of solids [8]. The development of a theoretical description of RHEED for an arbitrary surface still constitutes the subject matter of many recent investigations (e.g., [9,10]). This is because the scattering of electrons by surface atoms is complex and, in principle, multiple elastic and inelastic scattering events should be taken into account [11]. The patterns for ideally flat surfaces can be analyzed using dynamical diffraction theory, in which elastic multiple scattering is taken into account. Namely, the movement of electron waves in a crystal can be successfully described within the two-dimensional (2D) Bloch wave approach [8,11,12,13]. However, theoretical treatment of this kind for real samples is still incomplete. In analyses of many features observed in diffraction patterns, the use of single scattering theories has been continually unavoidable.

If only single scattering events are considered, then diffraction patterns can be found by direct summing of contributions coming from all atoms that form a particular structure. This is a basic concept of the kinematical approach [14,15]. Such treatment may be quite effective in use because, in principle, it enables the description even of quite irregular arrangements of atoms (for such cases, the use of multiple scattering theories is very difficult). Furthermore, if the focus is only on determining the basic geometrical relations for the observed patterns and not on the amplitudes of the features observed on the screen, a simple theoretical analysis can be carried out to find the conditions of the maxima for interfering waves. Consequently, the drawing of respective Ewald spheres can be applied [16,17,18,19]. Still, it is worth pointing out that there are some offshoots of such simplification. For example, for crystals with unit cells containing atoms of different types, partial cancelling of individual scattering contributions may happen and such effects cannot be examined within the approaches employing only geometrical analysis.

The characterization of solids with X-ray or electron diffraction methods using the concept of the Ewald construction is well known. It seems important to add that the concept of the Ewald sphere can be also employed in the area of photonics [20]. In any case, a good introductory depiction of the Ewald sphere and its derivation on the basis of a single scattering (kinematical) description is given in the book of Kittel [21]. However, recently Barbour [22] showed that its details can be demonstrated more efficiently by employing modern tools of three-dimensional (3D) computer graphics. He wrote a respective computer code to be applied for X-ray diffraction. In the current paper, we follow a similar route for the case of RHEED.

## 2. Materials and Methodology

All materials used for modern materials engineering, electronics, etc. need a crystallographic characterization. In this article, for simplicity, we discuss theoretical results for Ag and its fcc structure, and for Si and its diamond-type structure. However, we also show an example of interpretation of experimental data for complex oxides, which are the subject of increasing interest.

In RHEED experiments, electrons are scattered by atoms forming a crystal. Nowadays, diffraction patterns can be typically recorded with the help of CCD cameras. RHEED observations can be conducted both for low-index surfaces and vicinal surfaces (e.g., [23,24]). Information on the arrangement of atoms at the sample surface can be extracted by analyzing the diffraction patterns. If the surface is relatively flat, then the pattern can be predicted with the help of the Ewald construction for rods as shown in Figure 1. In the manuscript, the term “flat surface” is used to mean that the sample surface is without structural defects such as vacancies, adatoms or small islands at the crystal top. In other words, in such cases we assume a regular arrangement of atoms in planes near the crystal–vacuum interface.

However, in general, the Ewald construction can be carried out for reciprocal space points or for rods. Furthermore, for RHEED, the Ewald construction can be employed in different, equivalent ways. For example, in the paper of Mahan et al. [18], it is assumed that the initial point of the incident beam wave vector is attached to the zeroth rod. In our case, we use the vector terminal point. In fact, we follow concepts presented by Larsen et al. [17], but our screen is rotated by 180° to closely reproduce pulsed laser deposition conditions. Altogether, it seems helpful to briefly discuss all the important details of our approach.

In our considerations, the coordinate system used to describe the geometry of the measurement setup is fixed. We assume that the surface of the sample is always perpendicular to the *z*-axis. However, we admit that the sample can be rotated around this axis. Furthermore, the direction of the incident electron beam, in all considerations, is assumed to lie in the *xz*-plane (i.e., in the plane determined by the condition *y* = 0). The angle between the incident beam direction and the surface of the sample can be varied. In the case of RHEED, moving electrons are scattered by atoms much stronger than X-rays [25]. Moreover, RHEED experiments are typically conducted for small glancing angles, and therefore electron waves are diffracted mainly in a few atomic layers near the sample surface. For such a case, it is more appropriate to employ the two-dimensional reciprocal lattice than the three-dimensional to analyze diffraction patterns [8]. Subsequently, it is necessary to identify the points of intersection of the Ewald sphere with the reciprocal space rods perpendicular to the surface [17,19]. The computer program presented in Section 3 was developed to realize this automatically. The positions of the rods in the planes parallel to the surface are fixed accordingly to a coordinate set of the 2D reciprocal lattice with the primitive vectors $a^{*}$ and $b^{*}$ defined as follows [8]:(1)$$\{\begin{array}{c} a^{*}=\frac{2\pi \left(b\times \hat{z}\right)}{a\cdot \left(b\times \hat{z}\right)}\\ b^{*}=\frac{2\pi \left(\hat{z}\times a\right)}{b\cdot \left(\hat{z}\times a\right)} \end{array}\;.\;$$

In Equation (1), a and b are the primitive vectors of the surface direct lattice, and $\hat{z}$ is the unity vector of the *z*-axis. It can be easily checked that if a and b are taken in the form $a=\left(a_{x},a_{y},0\right)$ and $b=\left(b_{x},b_{y},0\right),$ then $a^{*}=\left(2\pi b_{y}/\Omega \;,-2\pi b_{x}/\Omega ,0\right)$ and $b^{*}=\left(-2\pi a_{y}/\Omega ,\;2\pi a_{x}/\Omega ,0\right)$, where Ω is determined as follows: $\Omega =a_{x}b_{y}-a_{y}b_{x}.$ For crystals with the face centered cubic (fcc) structure, it is convenient to assume that $a=\left(\sqrt{2}/2\right)c_{latt}\hat{x}$ and $b=\left(\sqrt{2}/2\right)c_{latt}\hat{y}$, if the (001) surface is being considered and the azimuth [110] is selected for the incident beam (see Figure 2). It is assumed here that $\hat{x}$ and $\hat{y}$ are the respective unit vectors and $c_{latt}$ is the crystal lattice parameter. The same definitions can be applied for crystals with the diamond-type structure.

Our approach (and the related program) makes it possible to obtain detailed quantitative information on lattice parameters and surface reconstructions. For surfaces with large terraces, information on the average size of terraces can also be obtained (in such analyses, however, reciprocal space rods with finite thickness must be considered).

It seems useful to describe briefly how the wavelength λ of incident electron waves is determined. For high-energy electrons, the following formula should be used [8,26]:(2)$$\lambda =\frac{h}{\sqrt{2m_{0}eV\left(1+\frac{eV}{2m_{0}c^{2}}\right)}}\;$$

In Equation (2), e is the elementary charge and V is the accelerating potential of the electron gun. Further, h is Planck’s constant, $m_{0}$ is the rest mass of the electron and c denotes the speed of light in vacuum. If we assume that V is expressed in volts and λ in angstroms, then we have the relation:(3)$$\lambda =\frac{12.3}{\sqrt{(V\left(1+0.978\times 10^{-6}V\right)\;}}\;$$

It is worth noting that the relations given by Equations (2) and (3) are displayed somehow incorrectly in the important paper of Mahan et al. [18]. Their formula, similar to Equation (2), does not contain 2 in the parenthesis. Subsequently, the formula displayed for determining λ in angstroms contains the value 1.95 instead of 0.978. For electron energy of 10 keV (used in Mahan et al. [18]), this leads to the appearance of a small error that can be ignored (see the discussion in Section 3.3). However, for higher electron energies, the use of improper formulas may be a source of discrepancies between experiment and theoretical predictions.

Until now we have assumed that reciprocal space rods are infinitely thin. However, for a relatively flat surface with some terraces, it is useful to consider rods with finite diameters [16]. Then, with the help of the Ewald construction, it is possible to predict elongated streaks at the screen.

## 3. Results and Discussion

### 3.1. An Algorithm

To predict RHEED patterns, some computations need to be carried out. They can be divided into three parts.

Part 1: A finite number of pairs $\left(x_{p}^{’},y_{p}^{’}\right)$ are determined. A single pair represents coordinates of a certain nodal point of the 2D surface reciprocal lattice. In this part of the algorithm, the possible rotation of the sample around the axis perpendicular to the surface is taken into account.

Part 2: The 2D reciprocal lattice is embedded in a 3D reciprocal space. To each nodal point considered in the previous part of the algorithm, an infinitely long rod is assigned. The rods are perpendicular to the x’y’-plane. Next, for each rod, its intersection points with the Ewald sphere are determined. The center of the sphere is located at the point (−Kcosθ, 0,   Ksinθ), where K is the magnitude of the wave vector of the incident electrons and θ is the angle between the sample surface and the incident beam. The Ewald sphere radius is equal to K. The intersection points, with their z’-coordinate not smaller than Ksinθ, are further used for determining positions of the spots at the screen for the beams formed due to electron diffraction occurring at the surface.

Part 3: It is necessary to find the components $k_{xp},k_{yp}$ and $k_{zp}$ of the wave vectors $K_{p},$ which define the directions of diffracted waves. This can be achieved by computing the coordinates of the intersection points; however, the coordinate system originated in the center of the Ewald sphere should be applied. Finally, after executing the respective operation of the projection, it is possible to find pairs $\left(y_{p},z_{p}\right)$ quantifying spot positions at the screen. Altogether, the following formulas are used in this part of the algorithm:(4)$$\{\begin{array}{c} k_{xp}=x_{p}^{’}+K\cos\theta \\ k_{yp}=y_{p}^{’}\\ k_{zp}=z_{p}^{’}-K\sin\theta \end{array}\;$$
and
(5)$$\{\begin{array}{c} y_{p}=\left(k_{yp}/k_{xp}\right)\;L\\ z_{p}=\left(k_{zp}/k_{xp}\right)\;L \end{array}.\;$$

In Equation (5), L is the distance between the sample and the screen. It should be added that the following relation is satisfied: $k_{xp}=\sqrt{\left(K^{2}-k_{yp}^{2}-k_{zp}^{2}\right)}.$ However, because $k_{yp}^{2}$ and $k_{zp}^{2}$ are much smaller than $K^{2}$, in practice it can be assumed that $k_{xp}\approx K$.

The description outlined above is valid for the case of infinitely thin rods. However, a similar algorithm can be applied for rods with finite diameters. Namely, for each reciprocal vector, it is necessary to introduce a set of pairs $\left(x_{p,c}^{’},y_{p,c}^{’}\right)$, being the coordinates of the number of the points lying on a circle with the center located at the terminal point of this vector. All other steps are identical to those for the case of infinitely thin rods.

However, the polycrystalline samples cannot be included in our analyses. To achieve this, it is necessary to apply the kinematical theory in which the summation of the scattering contributions from individual atoms is considered.

### 3.2. Features of the Software

A desktop application was developed with the use of the algorithm presented in Section 3.1. The code was prepared in the Java language with the help of the following open-source packages: OpenJDK-15.x [27] and JOGL 2.x [28]. Utilities of OpenGL libraries (available via the second package [28]) were included in the code and, subsequently, visualization with lighting and shadowing was possible. The purpose of the software was to describe the phenomenon of RHEED diffraction, creating and employing objects in a three-dimensional environment. The graphical user interface for the application was developed with the help of Java Swing/AWT libraries, provided together with the Open JDK package [27].

The current version of the program was developed for crystals having fcc or diamond-type structures, with the additional assumption that (001) planes are parallel to the sample surfaces. The exemplary analysis shown in this subsection was carried out for crystals composed of Ag atoms. A number of functionalities defined for the program by selecting different options from the menu, ticking some boxes at the computer screen, and pressing control keys at the keyboard are available for any user.

A snapshot of the application is displayed in Figure 3. Images from three virtual cameras are shown simultaneously at the screen. Two cameras at fixed positions show images of the RHEED pattern and the Ewald sphere viewed from the front. The third camera can be located at different positions. Respective enlarged images of this kind are shown in Figure 4 and Figure 5. A user can move this camera by using control keys at the keyboard. It is also possible to see the descriptions of the displayed objects (after pressing character “C” on the keyboard; such descriptions are shown in Figure 3).

Further, a user of the computer program can vary the direction of the incident beam by changing the glancing angle θ. The sample can also be virtually rotated, i.e., different values of the azimuthal angle ϕ can be selected. The values actually in use are displayed at the bottom of the computer screen.

The application enables the saving of diffraction spot coordinates in a text file as described in detail in the next subsection.

### 3.3. Detailed Diffraction Pattern

Using the software described in the previous subsection, a list of coordinates of the spots on the screen can be generated. In Figure 6, we show the list prepared in a format which is accepted by Gnuplot, a popular program freely available for drawing plots of different kinds [29]. The corresponding distribution of the RHEED spots displayed with the use of Gnuplot is presented in Figure 7. An output data list (see Figure 6) can be applied for strictly quantitative comparisons of simulation results with experimental data.

However, the primary goal of executing the simulations described here was to check the validity of our computer predictions by comparing them with the theoretical results of Mahan et al. [18]. Thus, our input data were taken similarly to one of the cases described in their paper. We considered an Si(001) surface, the energy of electrons was assumed to be equal to 10 keV, and the incident beam azimuth was set to be [110]. Further, we assumed that θ=0° and L=310 mm. The results displayed nominally and graphically in Figure 6 and Figure 7 are for spots from the first Laue zone.

Here it seems important to recall some details of the work of Mahan et al. [18]. Actually, these authors presented an extensive analysis of how to predict diffraction patterns in reflection geometry. They employed the Ewald construction with rods, but, in their case, the center of the Ewald sphere was located at the terminal point of the incident beam vector (we apply a different construction as mentioned in Section 2). Nevertheless, the predicted positions of diffraction spots at the screen should be identical for both these cases. There are two sources of possible differences in the results. First, Mahan et al. [18] carried out all analyses with three-digit precision only (for example 5.43/2 is rounded to 2.72 in their work), while in our case all computations are executed automatically with much higher precision (due to the use of the Java programming language). Second, the formula of Reference [18] for the determination of the electron wavelength contains an improper relativistic correction (see Section 2).

Altogether, our results agree well with those obtained by the authors of the aforementioned paper. In their case, the first Laue zone spots lie on a circle with a radius of about 77.9 mm. In our calculations, if we use Equation (3) after replacing 0.978 by 1.95 (this value was applied by Mahan et al. [18]), we obtain a circle radius equal to about 77.5 mm. If we apply Equation (3) straightforwardly (with 0.978), we obtain the value of about 77.6 mm (see Figure 6 and Figure 7). This means that for electron energy of 10 keV, the relativistic correction of the wavelength has very small effects. We can also conclude that the difference between our results and those of Mahan et al. [18] is approximately 0.5% which is fully acceptable if we take into account that they used three-digit precision in their considerations.

### 3.4. Comparison with the Experimental Data

Finally, we show an example of the interpretation of an experimental pattern, employing the approach and the related computer program described in the earlier parts of our article. As already mentioned, the program makes it possible to obtain lists with the precise positions of diffraction spots. Subsequently, quantitative comparisons of theoretical patterns with actual photographic images are possible. The experimental pattern displayed in Figure 8a for strontium titanate (denoted formally SrTiO3) was collected in a pulsed laser deposition chamber at the Academic Centre for Materials and Nanotechnology of the AGH University of Science and Technology in Kraków. The energy of electrons was set to be 20 keV. Strontium titanate samples can be used as substrates to prepare multilayered nanostructures [7]. It is possible to enforce the termination of SrTiO3 crystals with TiO2 domains at surfaces and, in fact, such a sample was examined by us. In general, it is known that strontium titanate has a perovskite structure. Accordingly, to apply our approach employing the Ewald construction, we used real space vectors a and b that satisfy the condition |a|=|b|=3.905 Å. The set of spots predicted theoretically for the [110] azimuth is shown in Figure 8b. The pattern was determined, assuming that the glancing angle value is equal to 2.7°. We can observe very good agreement between the experiment and theory.

## 4. Conclusions

It seems that permanent progress in computer science is indeed important for crystallography even in areas where our understanding of physical rules does not change significantly. In this paper, we argue that employing the tools of computer graphics may be beneficial in carrying out structural analyses. With such tools, diffraction experiments can be emulated thoroughly (as discussed by Barbour [22]), and it is possible to demonstrate basic concepts of the theoretical description of wave interference. It should be emphasized that our program employing 3D computer graphics does not actually offer new possibilities of structural analysis, other than those known from the paper by Mahan et al. [18]. However, we can execute such analyses in a more convenient and modern way.

## Figures and Tables

**Figure 1 materials-14-03056-f001:**
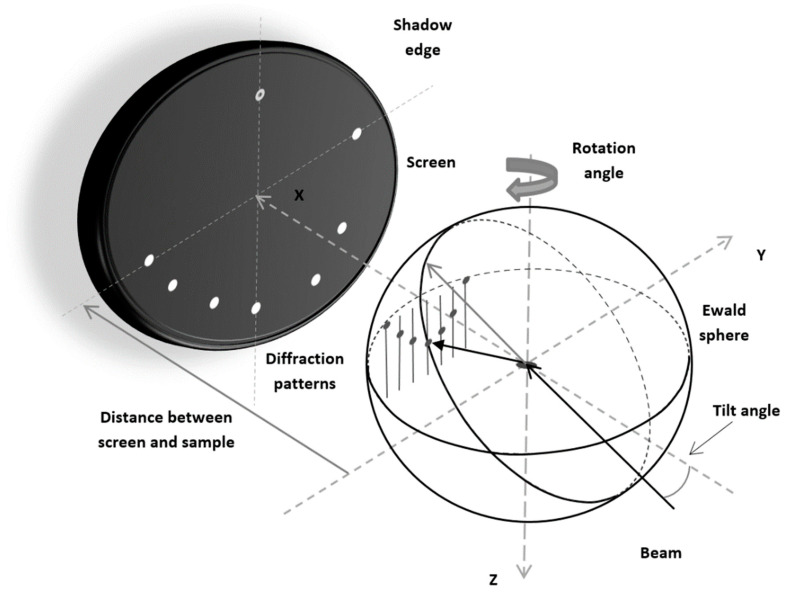
The Ewald construction for RHEED.

**Figure 2 materials-14-03056-f002:**
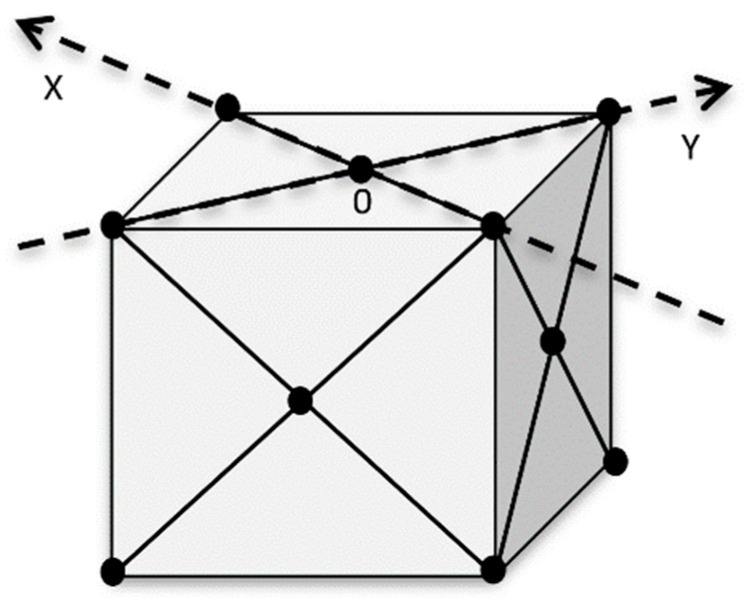
Model of the fcc structure as taken into consideration for the incident beam azimuth [110].

**Figure 3 materials-14-03056-f003:**
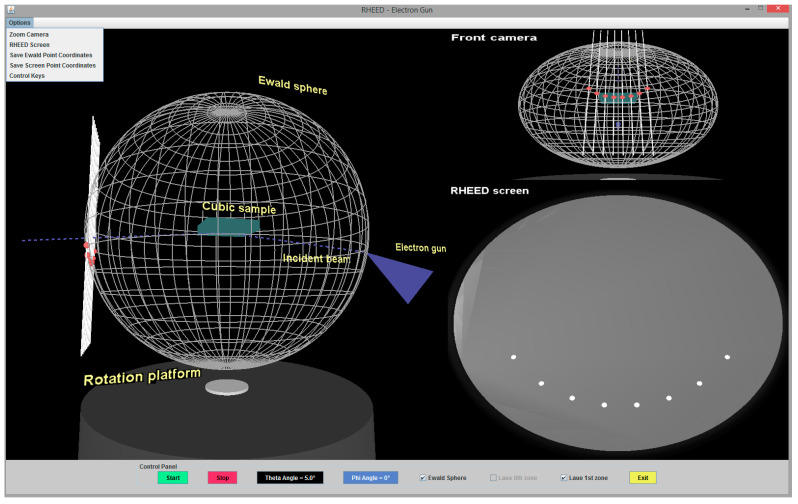
Snapshot of the application for drawing the Ewald sphere and finding the distribution of spots at the screen.

**Figure 4 materials-14-03056-f004:**
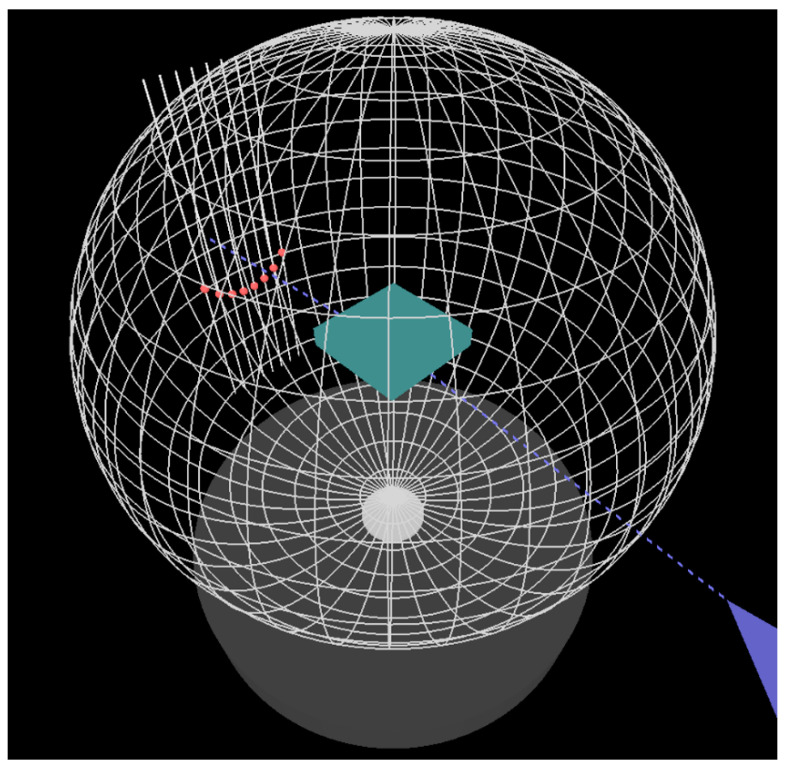
A skew view from a virtual camera that can be moved continuously to different observation points by a user of the application.

**Figure 5 materials-14-03056-f005:**
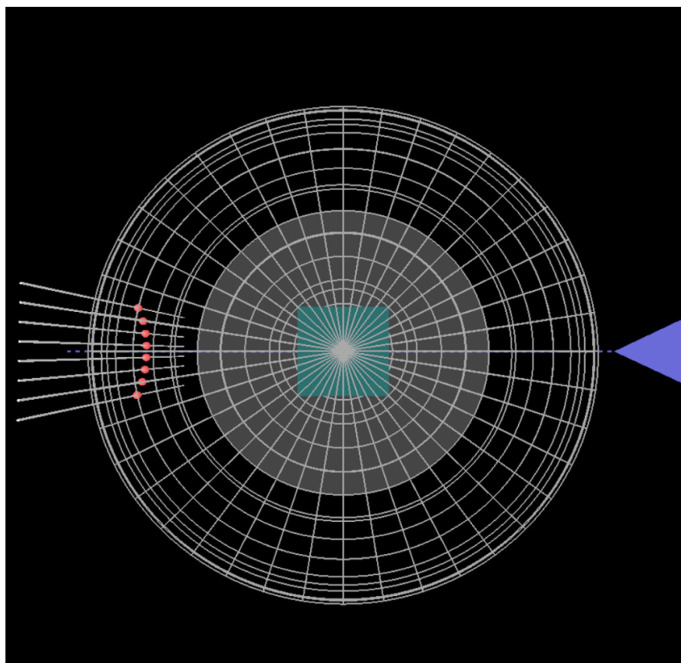
A top view from a virtual camera that can be moved continuously to different observation points by a user of the application.

**Figure 6 materials-14-03056-f006:**
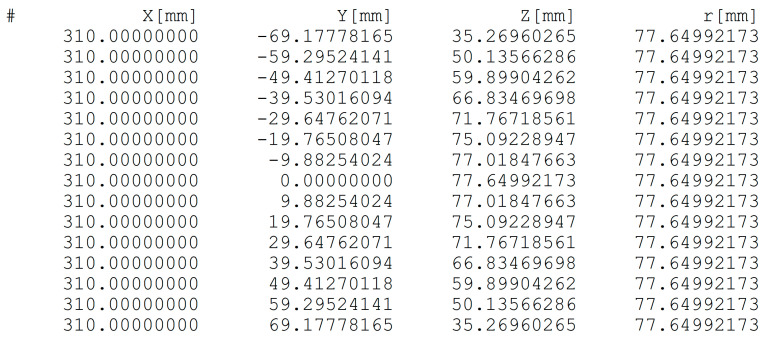
A list of coordinates of the spots at the screen generated by the software (additionally values of $\sqrt{Y^{2}+Z^{2}}$ are displayed).

**Figure 7 materials-14-03056-f007:**
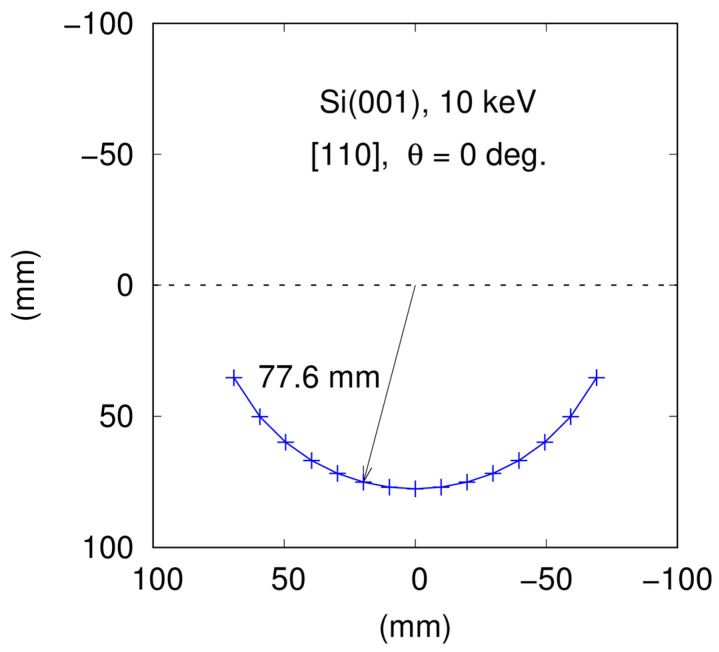
The distribution of the spots for Si(001) for the test data.

**Figure 8 materials-14-03056-f008:**
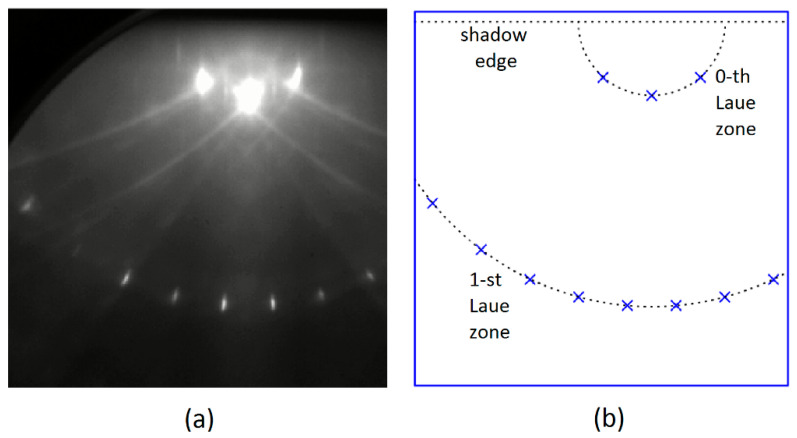
(**a**) Experimental RHEED pattern for a TiO2-terminated, SrTiO3(001) surface; (**b**) The pattern determined with the use of the Ewald geometrical construction.

## Data Availability

Data is contained within the article.

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
