# Peer review of "Simplified Determination of RHEED Patterns and Its Explanation Shown with the Use of 3D Computer Graphics"

_materials, 2021, doi:10.3390/ma14113056_

Round 1

Reviewer 1 Report

The study presented here concerns the determination of reflection high energy electron diffraction patterns through the development of 3D computer graphics. The manuscript is clear, well organized and enough details are given regarding the 3D computations.  However, after reading the text and the final conclusions, it appears that: 1) the precise scientific objectives of the authors together with the objectives compare to the state-of-the-art must be further detailed. What additional information does the 3D computation given here brings compared to a 2D model ? It is possible to extract more information or to add more precision regarding any possible surface features ? For instance such as strain relaxation, surface reconstruction, lattice parameter etc. ? Also compared to previous publications ?  2) what precise parameters can be extracted/determined ? Is it possible to extract quantitative parameters from this computation ? 3) the distribution of the spots given in figure 8 for the test of the data should be compared to experimental measurements ? 4) in the same figure, the glancing angle is set at 0° which is not adapted to experiments since it is always set above 0°. how will the distribution of spots vary as a function of the glancing angle ? why did the authors chose a value of 0° ?

Reviewer 2 Report

This paper is of interest to researchers using reflection high energy diffraction to monitor thin film growth and is recommended for publication.  This article is well written and clearly describes the scientific approach and results.  

Author Response

Thank you for these positive comments under this review. Please find the attached responses for reviewers 1 and 3.

Reviewer 3 Report

The manuscript from Kokosza et al. describes the utilization of computer graphics for the prediction of the high energy electron diffraction patterns via Ewald sphere construction. They well illustrated the program and the basic theoretical background behind it. They also compared their results with the literature, and seems reliable. In this manuscript, the authors provide a simple program to get the diffraction pattern from known crystalline which is useful and interesting. However, the scientific outcome is slightly weak. Can the program capable of dealing with the problem that for a given pattern interprets the real space structural information.

How about the other crystalline structures not only FCC or diamond structures, such as polycrystalline?

Can the diffraction strength be calculated from the Ewald sphere construction? The strength probably can be still related to the FT intensity of the real structure as in optics. (|DOI:10.1038/s41598-018-26119-8)

The author mentioned ‘flat surface, which is not clear.

The center of the Ewald sphere has a 0 y-axis value, is the illumination direction fixed in the x-y plane?

In Fig. 4, the author needs to briefly introduce the components, sample, stage, and detector positions or other information such as what are the blue dash line and triangle shape, etc? All this information is missed.

Round 2

Reviewer 1 Report

The authors have carefully answered the questions and the paper is therefore suitable for publication in Materials.

Reviewer 3 Report

Concerns are well addressed and the manuscript can be accepted.